# In situ and in vitro cryo-EM reveal structures of mycobacterial encapsulin assembly intermediates
Casper Berger [1,3,4] ✉, Chris Lewis [1,2,4], Ye Gao [1], Kèvin Knoops[1,2], Carmen López-Iglesias[1,2], Peter J. Peters [1] & Raimond B. G. Ravelli[1,5]

Prokaryotes rely on proteinaceous compartments such as encapsulin to isolate harmful reactions. Encapsulin are widely expressed by bacteria, including the *Mycobacteriaceae*, which include the human pathogens *Mycobacterium tuberculosis* and *Mycobacterium leprae*. Structures of fully assembled encapsulin shells have been determined for several species, but encapsulin assembly and cargo encapsulation are still poorly characterised, because of the absence of encapsulin structures in intermediate assembly states. We combine in situ and in vitro structural electron microscopy to show that encapsulins are dynamic assemblies with intermediate states of cargo encapsulation and shell assembly. Using cryo-focused ion beam (FIB) lamella preparation and cryo-electron tomography (CET), we directly visualise encapsulins in *Mycobacterium marinum*, and observed ribbon-like attachments to the shell, encapsulin shells with and without cargoes, and encapsulin shells in partially assembled states. In vitro cryo-electron microscopy (EM) single-particle analysis of the *Mycobacterium tuberculosis* encapsulin was used to obtain three structures of the encapsulin shell in intermediate states, as well as a 2.3 Å structure of the fully assembled shell. Based on the analysis of the intermediate encapsulin shell structures, we propose a model of encapsulin self-assembly via the pairwise addition of monomers.

Compartmentalisation of cellular processes is essential for all domains of life. Although the evolution of eukaryotes favoured the development of a complex endomembrane system (organelles) for compartmentalisation, relatively simple proteinaceous compartments are present in all domains of life. It is the complexity, rather than presence, of intracellular compartmentalisation that differentiates the eukaryotes from the bacteria and archaea[1]. Proteinaceous compartments are commonly found in bacteria and are collectively known as bacterial microcompartments (BMCs). They are characterised by a self-assembling icosahedral shell, but can be functionally distinct[2].

Encapsulins are a class of BMCs characterised by their conserved subunit (Enc), which self-assembles to encapsulate enzymatically active cargoes and can remain porous to cargo substrates. The number of Enc subunits assembled into the encapsulin shell corresponds with the triangulation (T) number of the icosahedron. Various triangulation numbers of encapsulin isolates have been reported, including $T = 1$ in *Thermotoga maritima*[3–5], *Mycobacterium smegmatis*[6], *Haliangium ochraceum*[7] and

*Mycolicibacterium hassiacum*[8], $T = 3$ in *Myxococcus xanthus*[9,10] and $T = 4$ in *Quasibacillus thermotolerans*[11]. The overall structure of Enc subunits across species with different T numbers is conserved and consists of a peripheral domain (P-domain), an axial domain (A-domain), and an extension loop (E-loop)[12]. There is, however, notable variation in E-loop length and angle that appears to be specific to the T number of the encapsulin shell[13]. Although 3D structures of fully assembled encapsulins have been well described and characterised, to our knowledge, little to no structural data is available on intermediates during encapsulin assembly, whether they are formed from monomers, dimers, trimers or pentamers, and how they incorporate their cargo. Native mass-spectrometry experiments showed that Enc dimers assemble into a rigid container[14]. Partially assembled states were observed as well, including 58- and 30-mers[14].

Encapsulins have sub-nanometre sized pores located at the points of five and threefold symmetry, and pores have been identified at points of twofold symmetry[6] and at the interface of two subunits[4,8] which are also known as two-fold adjacent pores[15]. The pores have variable diameters and

[1]Division of Nanoscopy, Maastricht Multimodal Molecular Imaging Institute, Maastricht University, Maastricht, The Netherlands. [2]Microscopy CORE Lab, FHML, Maastricht University, Maastricht, The Netherlands. [3]Present address: Structural Biology, The Rosalind Franklin Institute, Harwell Science & Innovation Campus, Didcot, United Kingdom. [4]These authors contributed equally: Casper Berger, Chris Lewis. [5]Deceased: Raimond Ravelli. ✉e-mail: casper.berger@rfi.ac.uk

charge; possibly allowing selective entry of cargo substrates[4,13,16]. Recent structural studies show structural variation in the diameter of the fivefold pore, which may suggest regulation of pore permeability[3,7,9,11,17]. The pores, however, are too small to allow entry of assembled cargo proteins. It is therefore most likely that cargo proteins are incorporated during the assembly process of the encapsulin shell[18,19]. The encapsulation and release of cargo seems therefore intrinsically linked to assembly and disassembly of the encapsulin shell, respectively. Recently, the structure of several cargo proteins within the encapsulin shell have been determined, including ferritin-like proteins[7,9], and dye-decolorising peroxidase (DyP)[6]. Assembly of Enc into the encapsulin shell can still occur without cargo proteins, as Enc assembles into complete encapsulin shells when independently expressed in mammalian cells or bacteria from other species[18,20]. Encapsulin cargo proteins are classified into core and secondary cargo, where core cargo genomic sequences are located on the encapsulin operon, usually directly upstream of Enc, and secondary cargo sequences are located elsewhere in the genome[21]. The loading of different cargo proteins into encapsulins is assisted by a cargo-loading-peptide (CLP) sequence, which is often at the C-terminal, and sometimes N-terminal of the cargo protein[21]. The CLP sequence can also be used to load exogenous cargoes into encapsulins[18,20], which, together with encapsulin self-assembly, enable a wide range of bioengineering applications[22–27].

Encapsulin functions are largely dependent on their cargo proteins. $T = 1$ encapsulins of *Mycobacterium tuberculosis*, the causative agent of tuberculosis, and the closely related *Mycobacterium marinum*, are known to encapsulate the DyP as a core cargo with iron storage ferritin protein (BfrB)[28] and FolB as possible secondary cargoes[6,8,29]. Encapsulin and their cargo proteins are known to be involved in intracellular survival of *M. tuberculosis* during infection by resisting oxidative stress, as knockouts of BfrA and BfrB, or DyP with Enc have been shown to reduce growth during infection of a guinea pig model or macrophages respectively[30,31].

The compartmentalised metabolic functions of encapsulins and their potential in bioengineering and biomedical research sparked a large variety of studies providing a wealth of biochemical and structural data of the fully assembled encapsulin shell. However, better sampling of the conformational landscape of encapsulin assemblies may help understand dynamic processes, such as shell assembly and cargo loading. While some recent structural studies were able to recapitulate aspects of structural diversity of the encapsulin shell, such as loaded cargo proteins and variable pore diameter[3,7,9,11], it remains largely unknown how full encapsulin shells are being built up. As with other proteins, encapsulins function in a complex and crowded cellular environment; isolation of macromolecules from their environment may cause loss and change of structural information[32–34]. Despite this, there is currently no structural information available for encapsulins in their native intracellular environment, where they perform their functions. Native intracellular structures may reveal binding partners lost upon isolation, intermediate shell assemblies and structural diversity in loaded cargoes in the cell.

Here, we study different structural states of Mycobacterial encapsulin complexes using a complementary in situ and in vitro approach. We employed cryo-focused ion beam (FIB) milling to prepare thin sections of rapidly frozen bacteria, both in isolation and in infection context, and investigated them using cryo-electron tomography (CET), enabling direct visualisation of encapsulin in its native environment. We show that in situ $T = 1$ encapsulins, although morphologically similar to their in vitro counterparts, frequently appear to have flexible attachments to their shell, here termed 'tails'. We present in situ encapsulins as a dynamic complex with and without cargoes in various assembly and disassembly states. We complement our in situ methodology with high-resolution in vitro single particle data obtained from a heterogeneous sample. We provide a 2.3 Å map of *M. tuberculosis* Enc from a fully assembled encapsulin, and reveal three stable intermediates of encapsulin assembly or disassembly in vitro. We compare the relative stability of Enc$_{tb}$ monomers in these intermediates to elucidate how the encapsulin shell may assemble. Based on these observations, we hypothesise a model of encapsulin shell assembly by the

addition of Enc dimers. These insights into encapsulin shell assembly can assist further studies into encapsulin intracellular functions, its role as a virulence factor and aid the development of encapsuling shells as a bioengineering tool.

## Results

### In situ *M. marinum* encapsulin structure

To directly visualise native encapsulin structure in its unperturbed cellular environment, we used CET on cryo-FIB-lamellae to visualise *M. marinum* grown in different culture media and in infected dendritic cells. From 217 tilt-series, we identified 439 intracellular encapsulins, both with and without infection contexts. The encapsulins appear to be randomly dispersed throughout the cell and do not colocalise with any cellular features or membranes (Fig. 1). However, they are excluded from nucleoids: large areas in the cell containing the genome, which are devoid of ribosomes[35–37]. No extracellular encapsulins could be found. Inspection of the individual intracellular encapsulins reveals structural features, cargoes, and intermediate states of assembly (Fig. 2). The majority of encapsulins were loaded with cargo (>90%), which are usually visible as heterogeneous densities. A distinct cargo protein could be identified in a small subset of encapsulins as a small ringed density offset from the centre of the encapsulin core, with an outer diameter of $6.9 \pm 0.4$ nm ($n = 13$ ring-shaped cargoes; Fig. 2). We observed that many intracellular encapsulins appear to have flexible, tail-like extensions attached to their shell exterior. Some encapsulin particles have a single tail ($n = 43$ encapsulin), others multiple ($n = 7$ encapsulin; Fig. 2). Most notably, in addition to tails and cargoes, we observed a variety of incomplete icosahedral encapsulin particles ($n = 40$ encapsulin; Fig. 2). We hypothesise that these are intermediate assembly or disassembly states. We also observed many partial encapsulins with cargoes inside ($n = 34$ encapsulin; Fig. 2).

### In vitro structure of the *M. tuberculosis* encapsulin

To study the structure of the encapsulin shell at near-atomic resolution, we overexpressed and isolated *M. tuberculosis* encapsulin from *E. coli*, which has 92.5% sequence identity (97% sequence similarity) with *M. marinum* (Supplementary Fig. 1). In addition, we studied partial encapsulin structures to characterise stable encapsulin shell intermediates in vitro, by using low pH conditions to disassemble encapsulin shells, followed by reassembly at neutral pH. This method had been previously used to encapsulate different endogenous as well as exogenous cargoes[16,19,38].

We used 89,072 particles which were found to be in the fully assembled state, yielding a 2.3 Å structure of the complete *M. tuberculosis* encapsulin shell (Fig. 3a-d; Supplementary Figs. 2 and 3; Table 1). The encapsulin shell forms a $T = 1$ icosahedron and has an inner diameter of 19 nm and outer diameter of 23 nm, measured along a fivefold symmetry axis. The icosahedral assembly of 60 copies of Enc$_{tb}$ can be described to consist of 12 pentamers, 20 trimers, or 30 dimers (Fig. 3e-g). The dimers are formed by the interaction of E-loops from Enc$_{tb}$ subunits of two neighbouring faces, forming a tight interaction. We used a model based on chemical thermodynamics[39] to predict the Gibbs free energy of Enc-Enc interactions, and found $\Delta G$ values of $-14.5$ kcal/mol for dimers, primarily mediated trough hydrogen bonds in the β-sheet formed between the E-loops, $-1.0$ kcal/mol for trimers and 0.8 kcal/mol for pentamers (Supplementary Fig. 4). This indicates that the formation of Enc dimerization via the E-loops is energetically highly favourable compared to Enc interactions within trimers and that Enc-Enc binding within pentamers would be energetically unfavourable. Next, we exploited the predictive power of Alphafold3[40] for protein multimer formation[41,42] to predict which Enc interfaces are favoured during complex formation and the structure for 2 to 10 copies of Enc$_{tb}$ were independently predicted based on the Enc$_{tb}$ sequence. Multimerization via the dimeric interface was consistently favoured over multimerization via the trimeric and pentameric interactions (Supplementary Fig. 5).

Enc$_{tb}$ has a strong structural resemblance to Enc of *M. hassiacum*[8] and *T. maritima*[4,5], which both also form $T = 1$ icosahedrons, with a root-mean-square deviation (RMSD) of all non-H atomic positions of 0.66 Å and

**Fig. 1 | Tomographic slices of *M. marinum* during intracellular infection and without an infection context.** *M. marinum* in the absence of an infection context (**a**) and during intracellular infection (**b**) expressing intracellular encapsulins (white arrows). Small irregular-shaped storage granules are predominantly present in bacteria without the infection context (black stars) whereas during intracellular infection larger spherical storage compartments that morphologically closely resemble lipid droplets in eukaryotic cells are predominantly present (white stars). Inner (red triangles) and outer membranes (white triangles) are clearly visible. During septum formation in cell division (black arrows) the outer membrane appears to remain continuous between the mother and daughter cell. Scale bars: 200 nm.

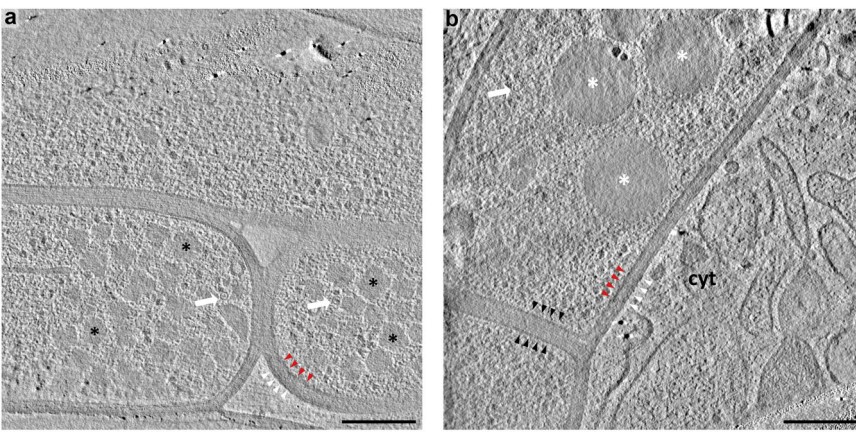

**Fig. 2 | Montage of tomographic slices of intracellular native encapsulin. a** Most encapsulins were found to contain cargo (>90%), which is most often observed as heterogeneous densities. **b** We identified a distinct circular cargo in some encapsulins (*n* = 13 encapsulin; yellow arrows). **c** We identified encapsulins with one (*n* = 43 encapsulin) or more (*n* = 7 encapsulin) heterogenous densities on the shell surfaces which we termed encapsulin tails (red arrows). **d** The shell of most encapsulins appears to be continuous, but some encapsulins appear to have a partially assembled shell (*n* = 40 encapsulin; cyan arrows). Most partially assembled encapsulins were still found to have cargoes contained inside (*n* = 34 encapsulin). Scalebar: 25 nm.

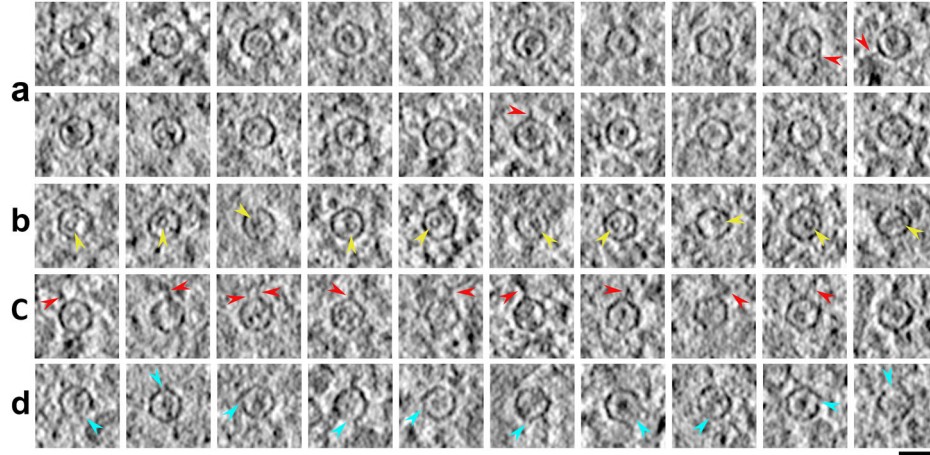

1.02 Å respectively. The most notable differences between $Enc_{tb}$ with Enc of other species are present in the A-domain, which forms the region near the fivefold pore and the E loop (Fig. 3h). Another noticeable difference is that the E-loop of Enc of T = 1 encapsulins is at a wider angle with the rest of the protein compared to T = 3 and T = 4 encapsulins (Fig. 3h). The E-loop also shows small differences compared to the E-loop of the T = 1 *T. maritima* encapsulins structure, including a rotation of the most distal section of it. Enc of *M. hassiacum* has a stabilising disulphide bridge between Cys134 and Cys254[8], which is absent for *M. tuberculosis* as it only has one cysteine residue(Cys254; position 134 is an alanine). T = 1 encapsulins appear to be more strongly conserved structurally than sequence-wise, as the sequence identity between $Enc_{tb}$ and *M. hassiacum* is 80.8% (91% sequence similarity), and only 35.2% (53% sequence similarity) between $Enc_{tb}$ and *T. maritima* (Supplementary Fig. 1).

The *M. tuberculosis* encapsulin shell has multiple different pores; single pores are present in the centres of each pentamer (9 Å diameter; Fig. 3a) and trimer (6 Å diameter; Fig. 3a) and two pores are present at the interface between two $Enc_{tb}$ subunits within the pentamer (9 Å and 5 Å diameter; Fig. 3a). We did not find a clear pore at the points of twofold symmetry. The diameter of the central fivefold pore is relatively large compared to the reported pore size of other species, such as 6.8 Å in *M. smegmatis*[6], and 8 Å in *M. hassiacum*[8]. The density map we obtained contains densities central in the 5-fold pores. Electrostatic surface colouring at neutral pH based on Coulombs law shows that the modelled pore has a slight positive charge on the outside of the shell and is neutral on the inside. The threefold pore has a slightly negative charge. The large pore at the interface between two $Enc_{tb}$ subunits has a negative charge while the smaller pore has a more neutral charge (Fig. 3i–k). Overall, the encapsulin shell structure of *M. tuberculosis* is

similar to T = 1 from other species, with some species-specific variations, such as the relatively large fivefold pore diameter and the rotation of the distal section of the E-loop.

## In vitro stable intermediates of the encapsulin shell

To explore the structural landscape of encapsulin shell formation, we used 2D and 3D (focused) classification and local symmetry to determine the structure of different stable encapsulin intermediates in an incomplete assembly state. This resulted in structures of three distinct intermediates of the encapsulin shell; a 48-mer (3,427 particles, 5.4 Å), a 52-mer (4,378 particles, 4.5 Å) and a 54-mer (6,294 particles, 4.6 Å; Fig. 4; Table 1; Supplementary Fig. 2; Supplementary Table 1-3). All three intermediates have an even number of monomers, appear to be closed on one side of the shell, and miss subunits on the other end. All three stable intermediates have an overall icosahedral structure with RMSD values for the 48,52 and 54-mer compared to the structure of the full shell of 1.08 Å, 0.60 Å and 2.66 Å respectively (Supplementary Movie 1-3), although the 54-mer is, compared to the full encapsulin, flattened in one direction and elongated perpendicularly to it (Supplementary Movie 4.). On the edges of the intermediate encapsulin shells, we observed many points of three and fivefold symmetry with one or more $Enc_{tb}$ subunits missing (Table 2). However, all dimers (in total 77) have both $Enc_{tb}$ subunits present, and where a subunit around five or three fold symmetry is missing, its dimeric partner is also missing (Table 2).

We found a local variation in how well the $Enc_{tb}$ subunits are defined. To determine which monomers are well resolved, we used a cross-correlation of the density map of the intermediate encapsulin structures with the rigid-body refined models of the partial

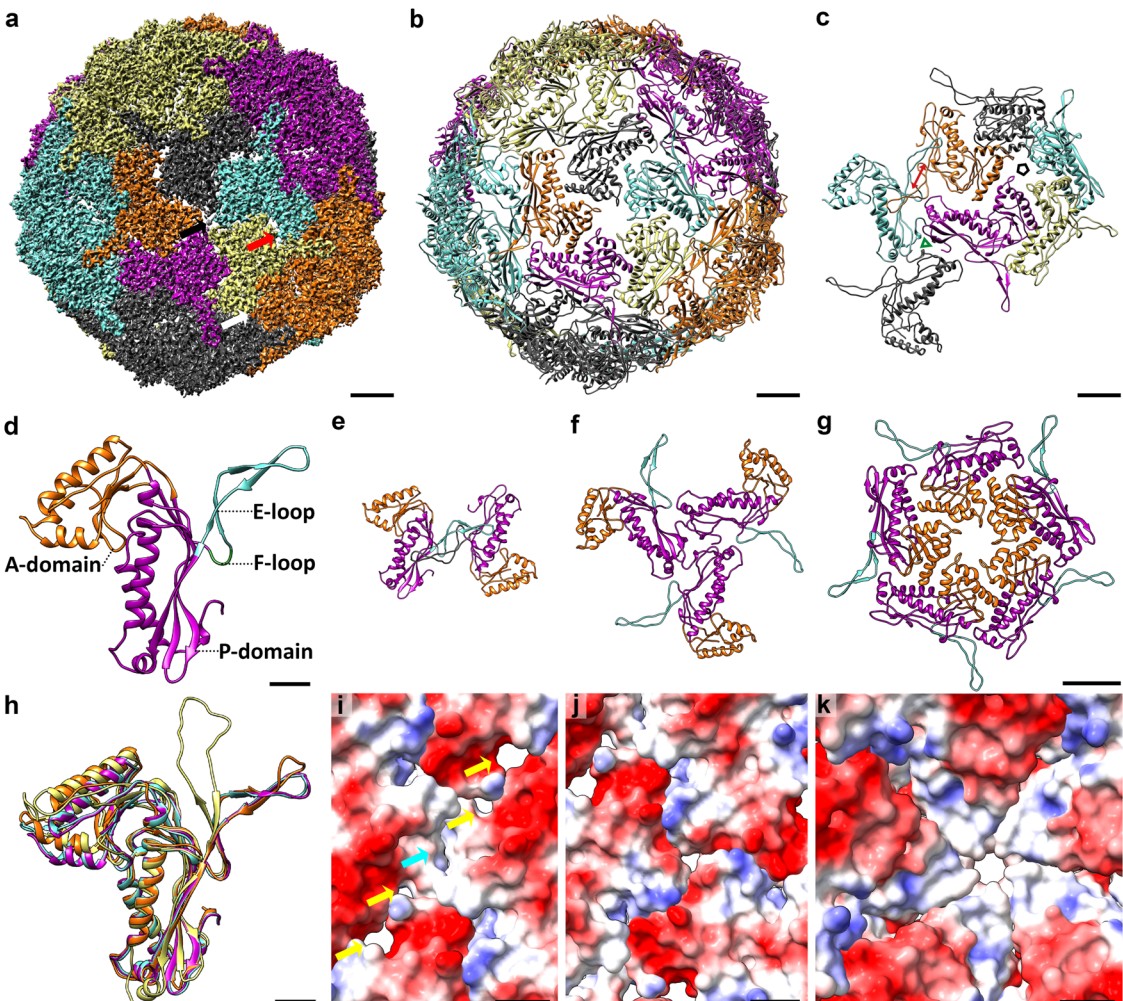

**Fig. 3 | Cryo-EM single particle structure of the encapsulin shell of *M. tuberculosis*. a** Electron-density map of the encapsulin shell. Each of the twelve pentamers consists of five Enc monomers (coloured orange, black, teal, yellow and purple in the central pentamer). The centre of each pentamer contains the fivefold symmetry pore (black arrow). Pores are also present at the points of threefold symmetry (white arrow) and on the interface between two Enc subunits (red arrow). Scalebar: 25 Å. **b** Atomic model ribbon view of the encapsulin shell of *M. tuberculosis*. scalebar: 25 Å. **c** Selection of the atomic model ribbon view in panel B to indicate the centre of the trimer (black pentagon), trimer (green triangle) and dimer (red double headed arrow). A total of 30 dimers, 20 trimers and 12 pentamers are present in the *T* = 1 encapsulin shell, where each Enc subunit is part of one dimer, trimer and pentamer. Scalebar: 25 Å. **d** Ribbon view of the Enc$_{tb}$ monomer with domains shown in different colours: P-domain (purple), A-domain (orange), E- loop (teal) and F-loop (green). Scalebar: 10 Å. **e–g** Ribbon view of the dimer, trimer and pentamer respectively with the domains shown in the same colour as panel D. Scalebar: 25 Å. **h** Overlay of the Enc monomers of *M. tuberculosis* (purple, this study, EMD-13164), *M. hassiacum* (teal, RCSB accession number 6I9G[8]), *T. Maritima* (orange, RCSB accession number 7K5W[5]) and *Q. thermotolerans* (yellow, (EMDB accession number 9383[11]). Scalebar: 10 Å. **i–k** Coulomb surface of the pores present on the interface between two subunits (yellow arrows) and on the points of three- and fivefold symmetry respectively. A point of twofold symmetry, where no clear pore is present, is indicated with a cyan arrow in panel (**i**). Positive charge is displayed in blue (10 kcal/(mol·e)) and negative charge in red (−10 kcal/(mol·e)). Scalebars: 10 Å.

encapsulin shells (Fig. 5a). The 48-mer has 39 fully resolved Enc$_{tb}$ subunits (defined as cross-correlation value > 0.75) and 9 are less well resolved (defined as a cross-correlation value 0.5 to 0.75; Fig. 5b and C, Table 3). The 52-mer consists of 39 fully resolved, and 13 less well resolved monomers, and the 54-mer consists of 27 fully resolved monomers and 27 less well resolved monomers (Table 3). Not surprisingly, we observed higher cross-correlation values for Enc$_{tb}$ monomers on the far side of the opening and lower cross-correlation values closer to the opening; in particular directly on the edges (Fig. 5d). The 48-mer and the 52-mer differ primarily by the addition of two more Enc$_{tb}$ dimers along the edge, around a point of threefold symmetry. The addition of these dimers appears to increase the stability of Enc$_{tb}$ subunits that previously were located directly on the edge of the hole (Fig. 5e). Summarising, the three intermediates were found to be largely icosahedral in structure with a single opening,

with more stable Enc subunits away from the opening. Partial pentamers and trimers were present in different configurations, whereas dimers were always complete.

## Discussion

In this work, we show evidence for structural heterogeneity in mycobacterial encapsulins using a combined in vivo and in vitro cryo-electron microscopy approach. Cryo-electron tomography on FIB-lamellae of *M. marinum*, with and without infection context, revealed encapsulins with heterogenous cargoes, partially assembled (or disassembled) encapsulins, and we observed flexible extensions on the exterior of the encapsulin shell not previously described, that we termed encapsulin tails. Single-particle analysis microscopy on heterogeneous preparations of isolated encapsulins, resulted in models for the structures of fully assembled encapsulin shell at 2.3 Å, and three structures of encapsulin in a partial state of assembly or disassembly at

**Table 1 | Cryo-EM Single Particle Analysis Data collection and modelling information**

|  | 60-mer | 48-mer | 52-mer | 54-mer |
|---|---|---|---|---|
| EM data collection/ processing |  |  |  |  |
| EMDB ID | EMD-13164 | EMD-51500 | EMD-52340 | EMD-52341 |
| Microscope | FEI Titan Krios |  |  |  |
| Acceleration voltage (KV) | 300 |  |  |  |
| Camera (mode) | GATAN K3 (counting) |  |  |  |
| Number of fractions | 40 |  |  |  |
| Exposure time (s) | 1.8 |  |  |  |
| Total exposure (e-/Å2) | 40 |  |  |  |
| Defocus range (μm) | 0.7–2 |  |  |  |
| Pixel size (Å) | 0.834 |  |  |  |
| Magnification | 163000 |  |  |  |
| Number of micrographs | 3827 |  |  |  |
| Initial reference model | 6I9G (PDB) | in silico | in silico | in silico |
| Number of particles (final map) | 34,427 | 3427 | 4378 | 6294 |
| Number of symmetry operators | 60 | 37 | 37 | 26 |
| Global resolution (Å) | 2.29 | 5.42 | 4.51 | 4.58 |
| Modelling |  |  |  |  |
| PDB ID | 7P1T | 9GOT | 9HQ7 | 9HQC |
| Bond lengths RMSZ | 0.37 | 0.37 | 0.37 | 0.37 |
| Bond Angles RMSZ | 0.57 | 0.57 | 0.57 | 0.57 |
| Clashscore | 2 | 7 | 6 | 5 |
| Ramachandron Favoured | 98% | 98% | 98% | 98% |
| Ramachandron Allowed | 2% | 2% | 2% | 2% |
| Ramachandron Outliers | 0 | 0 | 0 | 0 |
| Sidechain outliers | 1% | 1% | 1% | 1% |

Structure and modelling statistics for the encapsulin shell 60-mer, 48-mer, 52-mer and 54-mer obtained in this study.

targeting peptide between Enc subunits could play a key role here, as it is shared by all cargo proteins. Promotion of viral encapsulin assembly by binding to the nucleic acid genome is termed assisted assembly, which is a known mechanism in many viruses[44].

We observed flexible tail-like extensions on intracellular encapsulins, not present on isolates and, to our knowledge, not described in previous studies. Their absence on isolated encapsulin could suggest that these densities are transient and/or labile and are lost during the isolation process. Further experimental evidence is needed to determine which protein(s) could form these tail structures, and what could be their function(s). While most encapsulins in the cell appeared to be in a fully assembled state, we also observed partial encapsulin assemblies. Like the intermediates observed in vitro, the intermediate encapsulins in vivo also appeared to form an incomplete icosahedron. The heterogenous encapsulin cargoes and flexible tail densities on the shell exterior, combined with the observation of different assembly states both in situ and in vitro, suggest that encapsulation is a highly dynamic process within the cell.

The high-resolution structure of the encapsulin shell of *M. tuberculosis* obtained in vitro adds to the growing structural knowledge on encapsulins across different bacterial species. It is overall similar to T = 1 encapsulins from other species. A notable difference compared to *M. hassiacum* is the absence of a disulphide bridge between $Cys_{134}$ and $Cys_{254}$ in the *M. tuberculosis* encapsulin. This supports the hypothesis that the thermal stability of *M. hassiacum* encapsulin at high temperatures (65 °C) could be attributable to its disulphide bridge[8]. We found that the *M. tuberculosis* structure obtained in this study as well as the previously published *M. smegmatis* structure have relatively large pores at the points of fivefold symmetry compared to other encapsulins[6]. This may be caused by the different angle of the two alpha-helixes in the A-domain in mycobacteria compared to encapsulins from other species. A recently published X-ray crystallography structure of of $Enc_{tb}$ corresponds well between the structure obtained in this study, but some structural variation near the 5-fold pore is observed[45]. Both the Gibbs free energy prediction using the structure we obtained as well as Alphafold multimeric structure prediction using only the $Enc_{tb}$ sequence indicate that the dimeric interface is preferred in multimeric Enc interactions. It should be noted that these methods are limited by lack of consideration for possible co-factors, enzymes, or the reaction environment. The predicted preference for interactions via the dimeric interface are reflected in the three structures of intermediate encapsulin shells we obtained, where all 154 $Enc_{tb}$ subunits interacted via their dimeric interface, whereas for the trimers and pentamers, only 135 out of 154 and 110 out of 154 $Enc_{tb}$ subunits were present in trimers and pentamers with all subunits occupied respectively. Similar predicted Gibbs free energy values have previously been reported for EncA dimer interactions in *Myxococcus xanthus*[9]. A deeper understanding of encapsulin structure in relation to its properties can benefit the modification of encapsulins for bioengineering applications.

By temporarily lowering the pH, we were able to reconstruct three intermediate structures of encapsulin, enabling the structural study of encapsulin assembly and disassembly in isolates at a much higher resolution than in the cell. By comparing how well individual monomers in the partial shells were resolved, we were able to see which parts of the partial shell are more stable than others. Our intermediate structures indicate that the structure of larger partial shells is largely icosahedral, with some deviations observed in the 54-mer. A Recent study reports distorted T = 3 encapsulin shells of *Myxococcus xanthus* loaded with a non-native cargo[46]. All three structures of the encapsulin shell intermediates have an opening on only one side, which suggests that assembly occurs roughly from one end of the forming shell to the other. Monomers away from the edges of the partial shell were more stable than those at the edges, implicating that Enc is stabilised by the icosahedral shell assembly. A potential downside of this approach is the use of low pH to disassemble encapsulin shells, followed by reassembly at neutral pH, for which the assembly and disassembly mechanism could differ compared to intracellular conditions. However, the high resolution of the structures obtained in this study, and the high

around 5 Å resolution. Analysis of the occupancy of dimers, trimers and pentamers in the intermediate shell structures suggests that encapsulins are primarily assembled from Enc dimers, rather than trimers or pentamers.

We found that intracellular encapsulin have cargoes heterogenous in appearance, including distinct ring-shaped cargoes with an outer diameter of 6.9 nm and encapsulins that appear empty. The mycobacterial encapsulin cargoes BfrB, DyP and FolB all have a central ring shape, with outer diameters of 12 nm, 8 nm and 7 nm, respectively[6,28,43]. This makes DyP and FolB possible candidates for the cargoes identity, but based on its size and appearance alone we cannot unambiguously assign its identity.

The presence of partial encapsulin shells with cargoes inside supports the hypothesis that cargo proteins are loaded into the shell during assembly. Partial encapsulins with cargo inside were also observed by Snijder et al.[14] While Enc is well known to be able to self-assemble into encapsulins without the presence of cargo proteins, we primarily observed (>90%) fully assembled encapsulins with cargo in situ. This may suggest a regulatory mechanism to promote the encapsulation of cargo proteins. Many encapsulin and shell proteins are encoded on the same operon, suggesting that co-translational assembly could promote cargo encapsulation by locally increasing the concentration of shell and cargo proteins[13]. Another mechanism could be that binding of cargo proteins to Enc assists in the initiation of encapsulin shell assembly, although it should be noted that Enc can effectively assemble in the absence of cargoes as well. Binding of the

**Fig. 4 | Single particle structures of three stable encapsulin intermediates.** Single particle structures of (**a**) 48-mer (5.4 Å), (**b**) 52-mer (4.5 Å) and (**c**) A 54-mer (4.6 Å) shown from different perspectives. The 48-mer and 52-mer retain their icosahedral geometry, whereas the 54-mer has a flattened appearance. Scalebar: 25 Å.

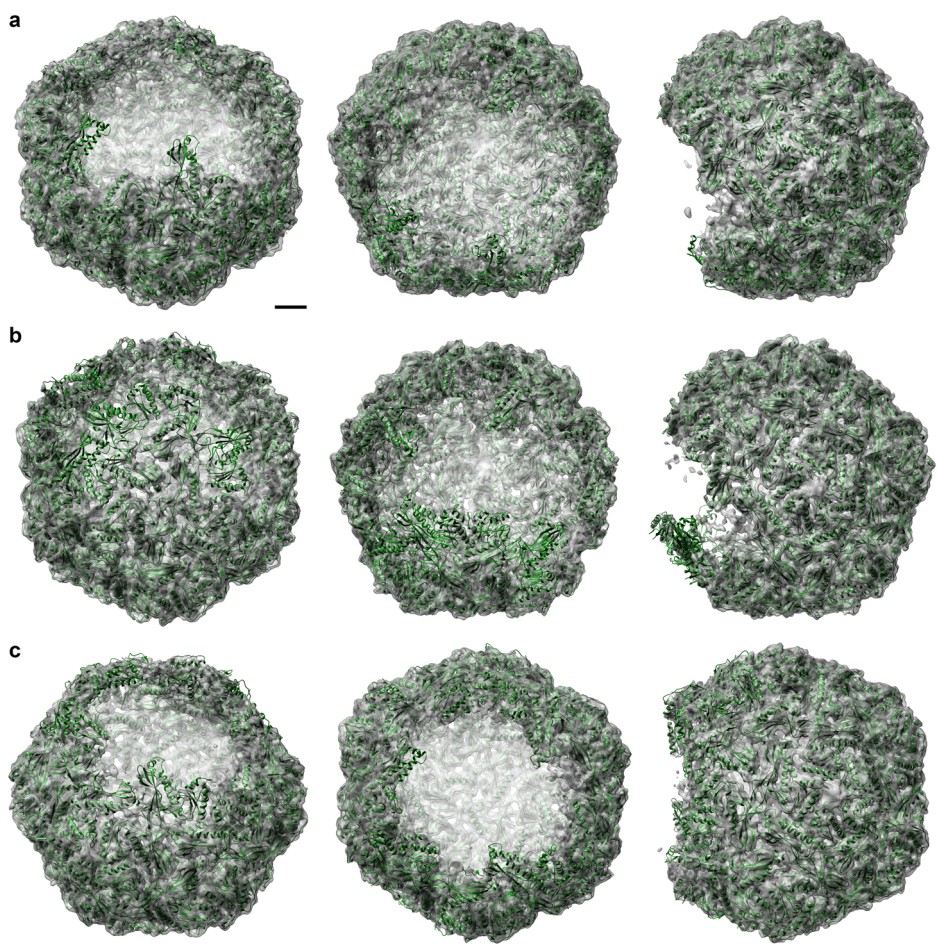

structural homology to encapsulin structures obtained without pH-dependant dis- and reassembly support the previous finding that this process is reversible[16,19,38].

Based on literature and our findings in this study, we propose a model for encapsulin assembly. A nucleation event initiates the formation of a multimer, from which the full encapsulin can start to assemble. Multiple copies of cargo assemblies have been observed in fully assembled encapsulins[6,7,9], suggesting that cargo proteins can become incorporated during the assembly process. Since the intermediate structures were open

from only one side, we reason that encapsulin are assembled largely from one side to the other, where the edge acts as a scaffold. Because all Enc$_{tb}$ monomers in the intermediate encapsulin structures are dimerised via the E-loops, we reason that the encapsulin shell is primarily assembled from dimers, or monomers that rapidly dimerise upon the addition of one monomer into the assembling shell. We consider assembly of the encapsulin shell from pre-assembled trimers and pentamers unlikely, because several trimers and pentamers with one or more missing subunit were observed in the encapsulin intermediates. The inclusion of Enc dimers into the assembling shell generally results in interactions at both trimer and pentamer Enc-Enc interfaces. Based on the predicted positive ΔG for Enc-Enc binding within pentamers, inclusion of dimers or pairs of monomers around trimers could be more prominent in driving these reactions. This would be consistent with the experimental structures obtained of 48- and 52-mer encapsulin shells, which differ by the addition of two dimers around a point of 3-fold symmetry (Fig. 5c) as well the multimer Alphafold predictions, which predicts the inclusion of dimers around points of threefold symmetry (Supplementary Fig. 5).

The pursuit of high-resolution structures creates the risk of ignoring more heterogenous structural states, which may biologically be highly relevant. The structural heterogeneity of encapsulins we observed in our in situ and in vitro data underlines the importance of studying dynamic protein assemblies, in isolation as well as in their native cellular context. Structural heterogeneity poses challenges in SPA processing. Tools have been developed to address such challenges[47–49], but these are not as widely adapted yet compared to the normal SPA workflow. While structures obtained with subtomogram averaging in situ *CET* are often still limited in resolution, ongoing developments[50–52] are likely to enable in situ structures at pseudo-atomic resolution scales[34,53,54] for a wider range of

### Table 2 | Subunit occupancy in stable encapsulin intermediates

| Number of monomers | subunits | 48-mer | 52-mer | 54-mer | total |
|---|---|---|---|---|---|
| Dimer | 1 | 0 | 0 | 0 | 0 |
| | 2 | 48 | 52 | 54 | 154 |
| Trimer | 1 | 4 | 3 | 0 | 7 |
| | 2 | 2 | 4 | 6 | 12 |
| | 3 | 42 | 45 | 48 | 135 |
| Pentamer | 1 | 0 | 0 | 0 | 0 |
| | 2 | 0 | 2 | 0 | 2 |
| | 3 | 6 | 3 | 9 | 18 |
| | 4 | 12 | 12 | 0 | 24 |
| | 5 | 30 | 35 | 45 | 110 |

Number of Enc$_{tb}$ subunits presents in the dimers, trimers and pentamers that are complete, or have one or more subunits missing, in the structures of the three stable encapsulin intermediates.

**Fig. 5 | Relative stability of $Enc_{tb}$ subunits along the edges of the encapsulin intermediates.** $Enc_{tb}$ subunits are coloured according to their cross-correlation with their density map. **a** Models and EM density maps of the 48-, 52- and 54-mers of the encapsulin shell shown with the open side facing forward. Scalebar: 25 Å. Examples of less well (**b**) and well resolved $Enc_{tb}$ monomers (**c**) overlaid with the density map. Scalebars: 10 Å. (**d**) Model of the 54-mer of the encapsuling shell shown from the side. Lower cross-correlation values are observed near the edge of the open side (Yellow and orange) whereas higher cross-correlation values are observed on the far-side (pink and purple). Scalebar: 25 Å. **e** Models of the 48- and 52-mers of the encapsulin shell. These intermediates primarily differ by the addition of two monomer pairs, which increases the cross-correlation values of monomers that were previously on the edge of the partial encapsulin (black arrows). Monomers shown in panel (**d**) and (**e**) are marked with grey dashed squares. Scalebar: 25 Å.

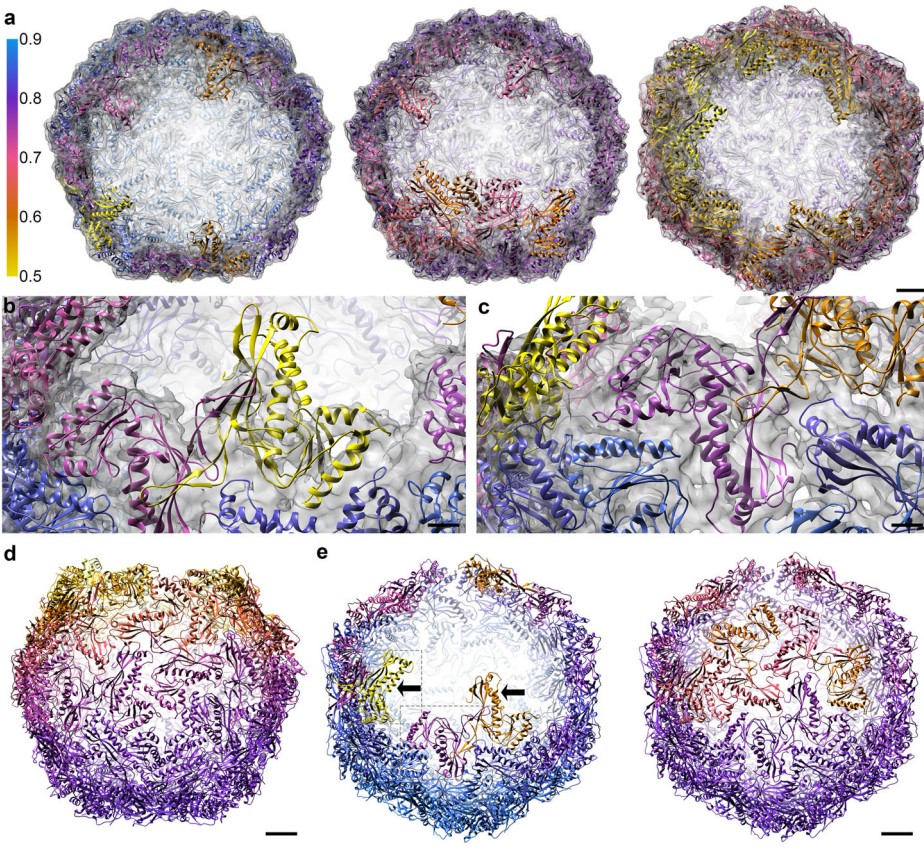

macromolecules, in their native setting. Our in situ work shows the presence of structural heterogeneity in a native context and our in vitro studies show that it is possible to obtain a molecular description of partial assemblies. Combined, these provide insight into encapsulin assembly mechanisms. Developments for improving SPA processing of structural heterogeneous samples, combined with the rapid pace of developments in in situ structural biology, harbour opportunities to understand the structural landscape of a wide range of macromolecules both in context of, and in relation to, cellular function.

## Methods
### Bacterial cultures for in situ studies
For all in situ work in this study, we used *M. marinum* over *M. tuberculosis* due to the lower biosafety requirements. Starter cultures of *M. marinum*, supplemented with 10% ADS and 0.05% tyloxapol, were grown in 7H9 medium whilst shaking at 200 rpm. Cultures were inoculated at an OD of 0.1 and grown for 5 days at 30 °C in 2 ml of medium (7H9 with 10% ADS, Sauton's medium or 7H9 with 10% OADC) without shaking. Bacteria were spun down in a 1.5 ml Eppendorf tube at 3000 x *g* for 1 minute and the supernatant was discarded. To pellet large clumps of bacteria, the initial pellet was gently resuspended in 100 µl of 7H9 medium with 10% dextran

before further centrifuge at 50 x *g* for 1 minute. 22 µl of the top of the bacteria suspension was transferred to an Eppendorf tube and 3 µl of protein-A gold was added just prior to grid preparation and sample vitrification.

To infect human dendritic cells, *M. marinum* USA was grown to an OD of 0.8 at 30 °C whilst shaking at 200 rpm in 20 ml of 7H9 medium supplemented with 10% ADS and 0.05% tyloxapol as described by van der Wel et al.[55] Initial cultures were spun down for 5 minutes at 5951 x *g* and the supernatant was discarded. The resuspended pellet (20 ml PBS) was spun down for 3 minutes at 358 x *g* and the supernatant was transferred to a Falcon tube for final centrifugation at 5951 x *g* for 5 minutes. The supernatant was discarded, and the pellet was resuspended in 1 ml of PBS.

### Monocyte isolation and differentiation
Blood monocytes were isolated from a Human blood buffy coat (Sanquin) with RosetteSep (Stemcell Technologies) as described by the manufacturer. Isolated monocytes were grown in AIM V medium (Thermo Fisher Scientific, cat. 12055083) at 37 °C and 5% $CO_2$ for 1 day in Petri dishes (Greiner, ref 633179). For differentiation into dendritic cells, monocytes were cultured for 6 days in AIM V medium with 200 ng/ml Granulocyte-macrophage colony-stimulating factor (GM-CSF; PeproTech, cat 300-03) and 25 ng/ml interleukin-4 (IL-4; PeproTech, cat 200-04).

### Infection and vitrification
After washing with RPMI 1640 medium (Thermo Fisher Scientific, cat. 11875093), cells were detached from the Petri dish by adding 2 ml of TripleE (Thermo Fisher Scientific cat. 12604013), incubating for 10 minutes followed by scraping. Eight UltrAuFoil 200 mesh R2/2 grids (Electron Microscopy Sciences) were placed in a 35 mm Petri dish (Greiner, ref 627160), seeded with 400,000 cells, and left for 1 h to adhere. Dendritic cells were infected with a multiplicity of infection of 20 and incubated at 35 °C for 16 h. One hour before vitrification[56] with a Vitrobot Mark IV modified with a jet vitrification module[57] 10 nm BSA gold fiducials were added.

### Table 3 | Three stable intermediate encapsulins

| Number of monomers | Number of fully resolved monomers | Number of less well resolved monomers |
| --- | --- | --- |
| 48 | 39 | 9 |
| 52 | 39 | 13 |
| 54 | 27 | 27 |

Number of fully defined monomers (CC > 0.75) and less well resolved monomers (CC between 0.5 and 0.75) observed in the three encapsulin intermediates.

## Cryo-FIB lamella fabrication

Lamellae were fabricated with a SCIOS FIB/SEM dualbeam (Thermo Fisher Scientific), upgraded to Aquilos specifications (Thermo Fisher Scientific). The grid was initially sputter-coated with platinum (6 seconds, 10 W, 600 V, 30 mA) and an SEM overview of the grid was acquired with MAPS software 3.1 (Thermo Fisher Scientific). An organometallic platinum layer was deposited on the grid at three different positions for 2–4 seconds before milling lamellae. Lamellae were milled with rectangular milling patterns or wedge pre-milling[58] with an initial width between 10 and 16 μm at 30 kV and a current between 0.05 and 1 nA, with a milling angle of 11° relative to the grid. To reduce crack formation in the lamellae, micro-expansion joints were created[59]. The finished lamellae were sputter-coated with platinum (3 seconds, 10 W, 600 V, 30 mA).

## Cryo-electron tomography

Tilt series were collected on a Titan Krios (Thermo Fisher Scientific) at 300 kV using a K2 camera (Gatan) operated in in electron counting mode (pixelsize 4.24 Å). Low-dose overviews were acquired at high defocus to navigate the lamella and to select sites for tilt-series acquisition. Tilt series were acquired using Tomography software package 4 (Thermo Fisher Scientific) with a total fluence of ~100 e-/Å$^2$ using a bidirectional tilt scheme between –50 and +50 degrees with 2-degree increments, corrected for the pre-tilt of the lamella. A total of 217 usable tilt series were acquired; 45 of infected dendritic cells, 66, 78, and 28 of cells in 7H9 10% ADS, Sauton's, and 7H9 OADC, respectively. The outer diameter of the ring-shaped cargo was measured with FIJI[60].

## In situ *data* processing and tomogram reconstruction

Alignment of dose fractions was done with MotionCor2 version 1.1[61] and tilt series were aligned with endocytic fiducial-based alignment[56] and reconstructed by weighted back-projection with IMOD software package 4.10.28[62] with fiducial-based local alignment and CTF correction by phase-flipping. For visualisation and particle picking, flipped and 4 x binned tomograms were filtered with TomDeconv EM[63] and encapsulins were manually picked. For the tomograms shown in Fig. 1 the platinum layer was computationally subtracted with Masktomrec[64] from 4 x binned tomograms. Encapsulin montages (Fig. 2) were prepared from 2 x binned and filtered tomograms.

## Protein expression and purification for in vitro studies

The codon optimised $Enc_{tb}$ operon of *M. tuberculosis* (Rv0798c) was synthesised (Eurofins Genomics) and cloned into the pRSET vector for transformation into *E. coli* C41 cells. Recombinant cells were cultured in LB at 37 °C with shaking at 200 rpm to an OD$_{600}$ of 0.5 and expression of Enc$_{tb}$ was initiated by adding 0.5 mM isopropyl-β-D-thiogalactoside (IPTG). After initiation, cultures were grown for 16 h at 18 °C with shaking at 150 rpm before harvesting cells by centrifugation at 4300 rpm for 30 min at 4 °C. Harvested cells were resuspended in 40 mL of buffer A (20 mM Tris 8.0, 300 mM NaCl), containing 2 U ml$^{-1}$ benzonase, and one protease inhibitor tablet (Roth) for subsequent lysis by sonication. Sonication was performed on ice with a total sonication time of 5 min, with pulses of 15 sec between 50 sec intervals. Cell debris was removed by centrifugation at 20,000 rpm for 30 min at 4 °C. After adding ammonium sulphate (~34% saturation), the supernatant was rotated for 1 h at 4 °C before centrifugation at 14,000 rpm for 20 min at 4 °C. The pellet was resuspended in buffer B (20 mM Tris 8.0 and 150 mM NaCl) and further purified by Superose 6 10/ 300 column (Sigma-Aldrich) pre-equilibrated with buffer B. Sample purity was assessed by SDS-PAGE on a 12% polyacrylamide gel. The sample was concentrated to 2.5 mg/ml using an Amicon Ultra-4 centrifugal filter with a 10 kD cut-off.

## Disassembly and assembly of Enc$_{tb}$ in the presence of BfrB

For disassembly and reassembly, isolated encapsulins were incubated at room temperature for 2 h at pH 3 by the addition of 0.1 M HCl. The solution was neutralised by the addition of 0.1 M NaOH. Enc$_{tb}$ was incubated for

18 h at room temperature to allow reassembly into encapsulin shells. Blue Native-PAGE was run using aliquots prior and after the disassembly/ assembly cycle, showing that both the disassembly and reassembly was incomplete.

## Single Particle data collection and processing

4 μL of reassembled Enc$_{tb}$ solution was applied to glow discharged (40 seconds) Quantifoil UltrAUFoil 300 mesh R1.2/1.3 grids before vitrification at 90% humidity and room temperature using a Vitrobot Mark IV (Thermo Fisher Scientific) set to blot force 5 and a blotting time of 3 seconds. A Titan Krios (Thermo Fisher Scientific) was used with a K3 direct electron detector (Gatan) operated in super-resolution counting mode and 40 fractions per movie using EPU (Thermo Fisher Scientific). Motion correction was done using MotionCor2[61] as directed by RELION 3.1.0[65] to yield 2 x binned dose-weighted micrographs of which the defocus was estimated using GctF[66].

All single particle analysis was done in RELION 3.1[65]. A full encapsulin shell template was created from manually picked encapsulin particles and used to auto-pick 89,072 particles from 3827 selected micrographs (Supplementary Fig. 3a) For the full encapsulin shell, any partially assembled encapsulins were discarded with initial 2D classification. Icosahedral symmetry was applied for subsequent masked (Supplementary Fig. 3c.i) 3D refinement. CTF refinement, Bayesian polishing and a final alignment resulted in a final post-processed full shell density map with a resolution of 2.3 Å (Supplementary Fig. 3a.i).

To autopick partially assembled encapsulins, we first created an in-silico partial encapsulin assembly model by removing monomers from our full shell model. A density map was produced from the in-silico model and then 12 2D projections were used as templates for particle picking (Supplementary Fig. 3b). After extraction of partial encapsulin assemblies, initial 2D and 3D classification using no reference mask and no symmetry resulted in two classes consisting of 6294 and 10,131 particles, respectively. After an initial 3D refinement without symmetry applied, both classes were further refined using a manually verified minimal set of icosahedral local symmetry operators. RELION's local symmetry function requires a binary softmap from which all symmetry translations and rotations are relative. To produce the binary softmaps (Supplementary Fig. 3c.ii), we fit a monomer from our complete encapsulin assembly model to the base of each partially assembled encapsulin shell in Chimera[67], created a 5 Å density map from the monomer model, mapped it to the 3D data array of the initial C1 3D refined partial assembly map, and softened the edges. To reliably establish which symmetry operators to use in further 3D refinement of the partial assemblies, preliminary partial assembly models were created by removing chains from the full shell assembly model in Chimera and then fitting the result to the C1 partial assembly maps. Aided with a manually verified lookup table (Supplementary table 1–3) to establish which RELION symmetry operation corresponds to which model chain, a minimal set of local symmetry operators was populated. The inclusion criterion for the minimal set of local symmetry operators was the full occupancy of the C1 partial assembly map around the corresponding monomer chain, as determined by local occupancy maps[68] (Supplementary Fig. 3a.ii, 3a.iii).

For all symmetrised 3D refinements of the partial assemblies, local symmetry was applied only where icosahedral transformations to the binary softmaps would encompass monomers with full occupancy. After each refine job, a new softmap was produced to account for slight shifts in alignments. For the initial class of 6,294 particles, 26 symmetry operators were used for refinement (27 monomers had full occupancy) of a partial encapsulin shell, which is modelled as a 54-mer (Supplementary Fig. 3a.vi) from a 4.6 Å density map. Inspection of the C1 map from the second 3D class (10131 particles) provided a minimal set of 37 local symmetry operators, which were used in a subsequent 3D refinement. Due to the presence of poorly defined densities situated away from the partially assembled encapsulin shell, we subtracted the projection[69] of the symmetrised part of the encapsulin shell from the experimental particles and focused the classification on the heterogeneous densities (Supplementary

Fig. 3c.iii). This provided two classes with 3427 and 4378 particles. The original particles in these classes were further refined with the non-subtracted region as a reference map (Supplementary Fig. 3c.iv) and the same minimal set of 37 symmetry operators used previously, resulting in a 48-mer (5.4 Å) and a 52-mer (4.5 Å), respectively (Supplementary Fig. 3a.iv, 1a.v).

For atomic model building of the full shell, the PDB model 6I9G[8] was fitted to the density and its sequence was adjusted into the one of *M. tuberculosis*. The icosahedral model was iteratively improved through rounds of manual adjustment in Coot[70], real-space refinement in Phenix[71] and structure validation using MolProbity[72]. For the model building of the partial shells, a cross-correlation of the monomer chains of the Enc structure determined from the full shell was performed against the maps of the partial shells. Cross-correlation values together with visual inspection in Chimera[67] were used to exclude monomers not present in the partial maps. To correct for local deviations in the partial encapsulin shell away from icosahedral symmetry, the remaining monomer chains were treated as rigid bodies in Phenix for a final model refinement with B-factor assignments per residue[71]. The momomers were assigned colours based on the average cross-correlation value of all residues with the partial shell map.

For predicting the Gibbs free energy of Enc-Enc interaction interfaces PDB files for a dimer, trimer and pentamer were uploaded to the PDBePISA server[39] without providing additional parameters, where for the trimer and pentamer the average solvation energy gain on complex formation for the trimer and pentamer interfaces was used. The Alphafold3 server was used to independently predict the structure of multimers with up to 10 copies of the amino acid sequence of Enc_{tb}. For sequence alignment, Enc sequences (uniprot-id's: O07181, Q9WZP2, K5BEG2, B2HH42) from different bacterial species were aligned using Clustal Omega[73] and visualised using ESPrit[74].

## Statistics and reproducibility
Tomograms were acquired from cells infected and vitrified in three independent experiments, and four for bacteria without an infection context. Enc_{tb} has been reproducibly expressed and purified in five independent experiments.

## Reporting summary
Further information on research design is available in the Nature Portfolio Reporting Summary linked to this article.

## Data availability
The density maps of the of the complete (EMD-13164) and stable intermediate encapsulin shells (EMD-51500; EMD-52340; EMD-52341), have been deposited in the Electron Microscopy Data bank (EMDB). Full shell (7P1T) and stable intermediate encapsulin models (9GOT; 9HQ7; 9HQC) have been deposited in the Protein Data Bank[75]. All raw data images used in all figures presented in the paper are available upon request.

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

## Acknowledgements
The UM acknowledge co-funding by the PPP Allowance made available by *Health~Holland*, Top Sector Life Sciences & Health, to stimulate public-private partnerships, under projectnr LHSM18067, as well as from the Netherlands Organisation for Scientific Research (NWO) in the framework of the National Roadmap NEMI project number 184.034.014. We thank the Microscopy CORE Lab members for their technical and logistics support, Christof Diebolder and Rebecca Dillard for their assistance with data acquisition with a Titan Krios at the Netherlands Centre for Electron Nanoscopy (NeCEN), and Matthias Müller for preliminary tomography data processing.

## Author contributions
C.B., C.L. and R.B.G.R conceived the study. C.B. performed sample preparation, lamella fabrication and tomography data acquisition and C.B. and C.L performed tomography data processing. Y.G. performed cloning, protein expression and isolation. C.L. and R.B.G.R. collected single-particle TEM data and performed data analysis and C.B., C.L. and R.B.G.R modelled the structure. C.B., C.L. and R.B.G.R wrote the manuscript and C.B. and C.L prepared the figures. C.B., C.L., Y.G., K.K., C.L.I., P.J.P and R.B.G.R. reviewed the data and the manuscript. R.B.G.R. supervised the project.

## Competing interests
The University of Maastricht has filed patents with RBGR CLP and PJP as inventors regarding sample preparation for cryo-EM. PJP is a shareholder of CryoSol-World. The other authors declare no competing interests.
