## [Transparent Peer Review file · Communications Biology]

In situ and in vitro cryo-EM reveal structures of mycobacterial encapsulin assembly intermediates

Corresponding Author: Dr Casper Berger

Version 0:

Reviewer comments:

Reviewer #1

(Remarks to the Author)

In their manuscript titled 'In situ and in vitro cryo-EM reveal structures of mycobacterial encapsulin assembly intermediates', the authors present in situ electron cryotomography and in vitro single-particle cryo-EM data on encapsulin assembly. Encapsulins form proteinaceous compartments that encapsulate protein cargoes, a process that is also essential for infection. The authors present low-resolution electron cryotomography data of such encapsulins in cryo-FIB tomograms of *M. marinum* cells within and outside of macrophages. The authors further assemble encapsulin shells of a highly related *M. tuberculosis* EncA encapsulin shell from recombinant protein in vitro, finding distinct assembly intermediates. This study is a useful addition to literature, expanding our structural knowledge and understanding of encapsulins, and I recommend its publication. I have, however, a number of points that I would ask to address.

Major points:

1. The resolution obtained from subtomogram averaging of the encapsulins in situ (~ 30 Å) is - as stated - relatively low, particularly given that the assembly is large and highly symmetric. The authors are furthermore not providing an FSC curves for the sub-tomogram average - please provide this. It would be helpful to discuss possible factors that limit resolution (partial assembly, flexibility, binding of additional factors, etc.).
2. The authors do not provide a table with data processing and model building statistics, as is customary when providing cryo-EM structures. Please provide this.
3. The authors partially base their interpretation that association of EncA into dimers is particularly favourable on a prediction of a change in Gibbs free energy, which suggests association into dimers is highly favourable (but significantly less so for trimers, and not for pentamers). While a reference to a manuscript termed 'Inference of Macromolecular Assemblies from Crystalline State' (Krissinel and Henrick, 2007 JMB) is given, it is unclear how these calculations were made and there is no mention in the method section describing them (or how this software may have been used). As far as this reviewer is aware, predicting the propensity of proteins to associate into multimers is very difficult. Hence, the authors would need to provide detailed descriptions on how these values were calculated, and discuss potential limitations of the method used. It may also be useful to comment on whether particularly strong interactions between the dimeric interface are present versus the trimeric and pentameric interfaces.
4. The authors create maps of several assembly intermediates of the *M. tuberculosis* encapsulin shell using single-particle cryo-EM. The methodology described here is somewhat confusing to me. I assume the authors processed the maps in RELION (as they mention they performed Motion Correction in RELION in the Methods section). They then create an average of the full capsid shell applying $T=1$ symmetry. The authors then mention employing 'local symmetry' to determine the structures of the intermediates. This part is unclear to me. Did the authors create manual symmetry operators to account for the subunits that are present in the respective intermediates as identified in C1 maps? If so, I find the term 'local symmetry' a bit confusing. It is also unclear which symmetry is applied for the subsequent classification. You mention you apply symmetry for 'focused classification' and then look at the resulting classes to determine occupancy of subunits. In RELION, the resulting density from Class3D would, however, be symmetrised, and hence it would be difficult to determine whether a particular subunit is present or not. To address this, please clarify the refinement/classification steps that were performed, in which order they were performed, and which kind of symmetry operators were applied during each, as well as which type of mask was used.

5. L442: The PDB map referenced in the manuscript '619G' does not exist - I assume the '1' is meant to be an 'l'? Also in line 509.

Minor points:

L98: The citations given do not specifically refer to encapsulin, although the placement of the citation suggests as much. Please rephrase sentence to 'isolation of macromolecules from their environment causes loss and change of structural information' if you would like to retain the citation.

L167: 'The density map we obtained contains protruding densities central in the 5-fold pores.' This phrasing is somewhat ambiguous as to whether the densities are within the pore or protruding from it (or both).

L201 onwards: You mention in the map that some subunits are 'partially resolved'. It would be useful to elaborate on what this means on the density level. Does the subunit smear out towards the edge? If so, to which extent, and is it a particular part of the subunit that seems to be poorly resolved? This would be especially helpful since you state that 'all EncA dimers were intact' (L198) but also say that an uneven number of monomers were only partially resolved. Perhaps an image of a well-resolved and partially resolved monomer would be helpful for the supplementary material.

L201: It would also be useful to provide overall RMSDs of the assembly intermediate models compared to the fully assembled shell to quantify the structural similarity between intermediates and the fully assembled shell.

L310: 'AI protein-folding' -> 'AI-based structural prediction'

L310: 'the AlphaFold Mania [...] has managed to address several of the initial limitations [...]' -> un-scientific, please rephrase

L314: '[...] use smartly chosen subsets [...]' -> colloquial, please rephrase

There is also a number of spelling and grammatical errors and inconsistencies in the manuscript, which distract a bit from the science. I point out the ones that I found below in a separate section but recommend to the authors to please double-check the manuscript.

Spelling errors:

L43: sub-unit -> subunit (spelled correctly in other places)

L124: do not colocalised -> do not colocalise

L143: 'we isolated E. coli overexpressed encapsulin from M. tuberculosis': this is grammatically ambiguous, better: 'we overexpressed and isolated M. tuberculosis encapsulin from E. coli'

L150: 'We, used 89.072 particles' -> 'We used 89,072 particles'. Also, it would be good to keep consistent within the manuscript whether or not you delimit large numbers (i.e., use a comma), which you sometimes do and sometimes do not.

L154: 'The icosahedral assemble of 60 copies of EncAtb, can be described to consists of [...]' -> 'The icosahedral assembly of 60 copies of EncAtb can be described to consist of [...]

L175: '(rmsd)' -> '(RMSD)'

L182: 'T1 encapsulins' -> 'T=1 encapsulins'

L239: 'as it's' -> 'as it is'

L262+264: 'M. Hassaicum' -> 'M. hassaicum'

L263: 'M. Tuberculosis' -> 'M. tuberculosis' (correct in other places)

L288: 'Because all EncA monomers in the intermediate encapsulin structure are present in a dimer' -> 'as a dimer'

L320: 'Developments for improving SPA processing of structural heterogeneous samples combined with the rapid pace of developments in in situ structural biology, harbour opportunities to understand the structural landscape of a wide range of macromolecules both in context of, and in relation to cellular function.' -> 'Developments for improving SPA processing of structural heterogeneous samples, combined with the rapid pace of developments in in situ structural biology, harbour opportunities to understand the structural landscape of a wide range of macromolecules both in context of, and in relation to, cellular function.'

L442: pdb -> PDB

L444: real space refinement -> real-space refinement

L448: cross correlation -> cross-correlation (correct in other instances)

L456: Clustal -> Clustal Omega

L491: 'We identified a distinct circular cargo in Some' -> 'in some'

L493: 'The shell of most encapsulins appears to be continues' -> 'continuous'

L497: Please have capitalisation after A), B) etc. consistent

L502: Selection of the Atomic model ribbon view -> atomic not capital

L508: Scalebar:25 Å -> space missing

L546: Supplementary figure 2: Please capitalise 'figure' to make consistent with S1

Reviewer #2

(Remarks to the Author)

In this paper, Berger et. al. showed in vitro structure of the fully assembled and three partially assembled encapsulin shells of Mycobacterium tuberculosis EncA. The maps obtained by cryoEM single particle analysis reached 2.4 Å resolution for the fully assembled shell, and 4 to 6 Å for the partially assembled shell. Comparing these structures, the authors proposed a model of encapsulin self-assembly from dimers and starting from one side of the shell to the other.

Berger et. al. also presented study of in situ encapsulin structures from cryo electron tomography (CET) on cryo-FIB-lamellae of Mycobacterium marinum, showing diverse encapsulins as complete or incomplete icosahedral particles, with various cargo presence and type. Flexible ribbon-like attachments on the exterior of the shell were observed, though their identity

and/or function are unknown. Subtomogram-averaging of the encapsulin particles in situ results in a low resolution 3 nm map, possibly because: 1. the magnification used in CET is low with pixel size 4.24 Å; 2. the tilt scheme used is the traditional bi-direction, instead of dose-symmetric; and 3. low number of total particles for the average.

Minor points:

1. Line 243 – 248 on page 8: “We observed flexible tail-like extensions on isolated encapsulins, not present on isolates and, to our knowledge, not described in previous studies. This raises the question which protein(s) could form these tail structures, and what could be their function. We consider it unlikely that these extensions are formed from EncA subunits in the process of assembly or disassembly, based on their flexible appearance, their appearance on fully assembled encapsulins, different contrast from the encapsulin shell and that these extensions have not previously been observed in isolated encapsulins.”

- a. Were the tail-like extensions observed on isolated encapsulins or not? The description here is inconsistent.
- b. The reasons why it is not from EncA subunits in the process of assembly or disassembly are unclear or weak:
 - i. What does “flexible appearance” mean? In terms of length, shape, or contrast?
 - ii. Were they not observed on partially assembled encapsulins? It seems that some are present in Figure 2D, e.g. in the 2nd, 4th, 6th images (please see attached screenshot).
 - iii. The “tail” could be a totally different structure made of EncA subunits, compared to the icosahedral shell structure, hence the difference in contrast or image intensity.
 - iv. The loss and change of structural information in isolated proteins could explain why the extensions were never observed previously in vitro, even if the extensions were made of the same EncA subunits.

2. Line 310 – 316 on page 10: “Over the last year, the AlphaFold Mania (Callaway, 2022) has managed to address several of the initial limitations listed in July 2021; the simultaneous design of sequence and structure (Anishchenko et al., 2021; Lovelock et al., 2022; Lutowski et al., 2022), enrichment with ligands and co-factors (Hekkelman et al., 2021), Small (Park et al., 2021) as well as ultra-large super- molecular complexes (Eisenstein, 2021; Callaway, 2022; Mosalaganti et al., 2022) and to use smartly chosen subsets of the PDB to train neural networks to predict distinct conformational states of transporters and receptors (del Alamo et al., 2022).”

This account of recent developments in using machine learning in protein structure prediction seems irrelevant and distracting to the points of the paper.

Reviewer #3

(Remarks to the Author)

Summary and General Assessment:

In the manuscript “In situ and in vitro cryo-EM reveal structures of mycobacterial encapsulin assembly intermediates”, the authors use cryo-ET and cryo-EM approaches to investigate the assembly dynamics and intermediate states of Mycobacterial encapsulin shells.

This study tackles an interesting and challenging problem regarding the assembly mechanism of encapsulin protein shells which has remained mostly elusive up till now. Some novel and interesting data are presented to support the hypothesis that icosahedral shell formation proceeds via the successive addition of shell protein dimers. However, it appears that the authors are not that familiar with the encapsulin field in general and make some inferences and statements that need to be corrected. Further, a number of inferences and statements are at best speculative and not supported by enough solid data. I am suggesting a number of experiments and changes to the interpretation/text that should be strongly considered by the authors. After all the points listed below have been addressed, I can recommend this study for publication in *Communications Biology*.

- The authors seem to refer to any encapsulin shell protein as EncA. This is incorrect, EncA specifically refer to the encapsulin shell protein encoded in the *Mycobacterium xanthus* FLP encapsulin operon.
- Page 3, line 82: This statement is most likely incorrect. BfrB and FolB were shown to be to some extent encapsulated upon heterologous overexpression with the encapsulin shell protein. However, their purported targeting peptides (TPs) are most likely not true TPs but simply C-terminal extensions very often found in proteins as they deviate substantially from all other known TPs. Further, they are not part of the respective encapsulin operon. In addition, native isolation of the highly homologous *Mycobacterium smegmatis* DyP encapsulin definitively showed that only the expected DyP cargo encoded in the encapsulin operon is encapsulated, not BfrB or FolB.
- Page 4, line 123: what is the observed average number and range of encapsulins per cell?
- Page 4, line 133: Again, FolB is not a well-known encapsulin cargo. The observed cargo is very likely the co-encoded DyP found in almost all Mycobacterial encapsulin systems. DyP forms a ring-shaped hexamer of the observed dimensions.
- Page 4, line 135: what does “many” mean, how many precisely? Could this be an artifact? Stating that it protrudes from the center of the pentameric vertex is simply speculation and cannot be stated as such based on the presented data/images. Were any of these “tails” visible after subtomogram averaging/sorting?
- Page 5, line 146: It is very difficult to draw any conclusions from an experiment like this carried out under non-physiological conditions (pH 3). It is certainly conceivable, if not likely, that the low pH might have altered the assembly mechanism. This should be taken into account, clearly pointed out, and discussed. This of course means that all data based on this experiment has to be considered in the context of non-physiological conditions. Any statements about assembly mechanisms or intermediates need to be qualified with this in mind.
- The authors should carry out SEC and DLS analysis of low pH samples, the proposed mechanism – based on dimer addition to growing shells – implies that encapsulin shell dimers should be present in solution which should be detectable

using the methods mentioned above.

- Page 6, line 170: What is shown in panel (I) of Fig. 3 is not 2-fold symmetrical and does not look like the 2-fold pore? Please clarify.

- Page 6, line 190: More details should be provided in the main text on how this analysis was done, also, supporting 2D and 3D classes should be shown in the context of the overall workflow that resulted in the identification of these heterogeneous shells.

- Page 7, line 220: I don't think this statement is supported by the data. What different cargos? Are the flexible extensions/tails real?

- Page 8, line 228: Again, the authors say "diverse" cargos, but do not present any evidence for the presence of diverse cargos, also, it is most certainly incorrect to state that you observed FoIB, it is most likely DyP, the main cargo of the system, however, you really cannot make any definitive statement here based on the available evidence.

- If the authors want to make more definitive statements about cargo identity, they should purify the native encapsulin from *M. marinum*, this would allow the definitive identification of all cargo(s) present under the given growth conditions.

- Page 8, line 238: This seems unlikely considering that all so far characterized encapsulin shells seem to very efficiently assemble even in the absence of cargo.

- Page 10, line 298: This whole paragraph seems unnecessary and disconnected from the rest of the manuscript,

Version 1:

Reviewer comments:

Reviewer #1

(Remarks to the Author)

I agree with the authors' decision to remove the STA density under the given circumstances. The description of image processing and model building has significantly improved and the new movies are a useful addition. The authors have addressed my comments. I recommend the manuscript for publication.

Reviewer #2

(Remarks to the Author)

Berger et. al had addressed previous reviewers' comments adequately. I recommend its publication.

Reviewer #3

(Remarks to the Author)

The authors have done a good job addressing all my original points and I can now recommend this manuscript for publication.

Reviewer #1 (Remarks to the Author):

In their manuscript titled 'In situ and in vitro cryo-EM reveal structures of mycobacterial encapsulin assembly intermediates', the authors present in situ electron cryotomography and in vitro single-particle cryo-EM data on encapsulin assembly. Encapsulins form proteinaceous compartments that encapsulate protein cargoes, a process that is also essential for infection. The authors present low-resolution electron cryotomography data of such encapsulins in cryo-FIB tomograms of *M. marinum* cells within and outside of macrophages. The authors further assemble encapsulin shells of a highly related *M. tuberculosis* EncA encapsulin shell from recombinant protein in vitro, finding distinct assembly intermediates. This study is a useful addition to literature, expanding our structural knowledge and understanding of encapsulins, and I recommend its publication. I have, however, a number of points that I would ask to address.

We thank reviewer #1 for their comments and recommending the manuscript for publication.

Major points:

1. The resolution obtained from subtomogram averaging of the encapsulins in situ (~30 Å) is - as stated - relatively low, particularly given that the assembly is large and highly symmetric. The authors are furthermore not providing an FSC curves for the sub-tomogram average - please provide this. It would be helpful to discuss possible factors that limit resolution (partial assembly, flexibility, binding of additional factors, etc.).

The resolution we obtained for the subtomogram average is indeed lower than what we had expected considering the icosahedral symmetry. We tried STA using both Relion and Dynamo independently and obtained maps similar in quality and resolution. We suspect that the initial alignment without applying any symmetry (with only 265 particles after classification) may not be good enough yet to fully benefit from the icosahedral symmetry. A contributing factor to this may be that at 2-4 nm resolution there appear to be few defining features to drive the alignment (see the figure below). We suspect that more particles are needed to overcome this initial barrier before fully benefitting from the icosahedral symmetry.

Original EM map obtained in this study lowpass filtered to different resolutions and displayed as a surface (top row) or as a central slice (bottom row). From left to right: original map, 5 Å filtered, 10 Å filtered, 20 Å filtered, 40 Å filtered. Scalebars: 10 nm.

We agree that an FSC curve should be provided with an STA density map. We tried to provide one, but this proved to be difficult as some required intermediate files have been lost. As none of the findings in this manuscript depend on this low-resolution density map, we decided that removing it from the supplementary figures was the best option.

2. The authors do not provide a table with data processing and model building statistics, as is customary when providing cryo-EM structures. Please provide this.

We added a table on data processing and model building statistics to the manuscript (Supplementary table 1).

3. The authors partially base their interpretation that association of EncA into dimers is particularly favourable on a prediction of a change in Gibbs free energy, which suggests association into dimers is highly favourable (but

significantly less so for trimers, and not for pentamers). While a reference to a manuscript termed 'Inference of Macromolecular Assemblies from Crystalline State' (Krissinel and Henrick, 2007 JMB) is given, it is unclear how these calculations were made and there is no mention in the method section describing them (or how this software may have been used).

We added a description of how the PDBePISA server was used to predict the change in Gibbs free energy for the dimer, trimer and pentamer interfaces to the materials and methods section. While rerunning the analysis to write this description we noticed a rounding error for the results of the trimer interface (-0.96 kcal/mol, which was cutoff to -0.9 rather than rounded to -1.0), which we corrected in the manuscript.

We also supplemented these results by using AlphaFold multimer prediction for different Enc_{tb} multimer conformations, which consistently favours binding via dimeric interfaces over the trimeric and pentameric interfaces. These results have been added to the results section and as a new supplementary figure (Supplementary Figure 5).

As far as this reviewer is aware, predicting the propensity of proteins to associate into multimers is very difficult. Hence, the authors would need to provide detailed descriptions on how these values were calculated, and discuss potential limitations of the method used.

We added a section to the discussion to discuss possible limitations of this prediction method for multimeric complexes:

“Both the Gibbs free energy prediction using the structure we obtained as well as deep-learning-based multimeric structure prediction using only the Enc_{tb} sequence indicate that the dimeric interface is preferred in multimeric Enc interactions. It should be noted that these methods are limited by lack of consideration for possible co-factors, enzymes, or the reaction environment.

It may also be useful to comment on whether particularly strong interactions between the dimeric interface are present versus the trimeric and pentameric interfaces.

We added a brief explanation on why the dimeric interface is predicted to be stronger in the results section and we visualised the predicted bonds together with the bond distances in a new supplementary figure (Supplementary figure 4).

4. The authors create maps of several assembly intermediates of the *M. tuberculosis* capsid shell using single-particle cryo-EM. The methodology described here is somewhat confusing to me. I assume the authors processed the maps in RELION (as they mention they performed Motion Correction in RELION in the Methods section). They then create an average of the full capsid shell applying T=1 symmetry. The authors then mention employing 'local symmetry' to determine the structures of the intermediates. This part is unclear to me. Did the authors create manual symmetry operators to account for the subunits that are present in the respective intermediates as identified in C1 maps? If so, I find the term 'local symmetry' a bit confusing. It is also unclear which symmetry is applied for the subsequent classification. You mention you apply symmetry for 'focused classification' and then look at the resulting classes to determine occupancy of subunits. In RELION, the resulting density from Class3D would, however, be symmetrised, and hence it would be difficult to determine whether a particular subunit is present or not. To address this, please clarify the refinement/classification steps that were performed, in which order they were performed, and which kind of symmetry operators were applied during each, as well as which type of mask was used.

We agree that incorrectly applying (local) symmetry operators creates the risk of resolving densities that are not actually present in the data. We therefore took several steps during processing to ensure that this would not happen. The initial classification steps and 3D refinement for the partial encapsulin structures were done without any symmetry or masks applied. Where local symmetry in later steps was applied cautiously by using a minimal set of symmetry operators for well-resolved Enc monomers only, away from the edges of the shell. All densities resolved near the edges of the partial encapsulin structure are therefore only aligned based on the local symmetry operators, but not averaged, so any signal in these areas is locally present in the raw data. We use phrasing for local symmetry as described in the Relion documentation: https://www3.mrc-lmb.cam.ac.uk/relion//index.php?title=Local_symmetry.

We extended the materials and methods section to provide much more details on how and when all the classification and focussed classification steps were done to obtain the partial encapsulin density maps. Where masks were (or were not used) we have explicitly stated it.

We also added a new supplementary figure to visualise the data processing steps (Sup. Fig. 3) as well as a supplementary table with processing and modelling information (Supplementary Table 1) and a supplementary table describing the exact symmetry operators used for the 48, 52 and 54-mer (Supplementary table 2).

5. L442: The PDB map referenced in the manuscript '619G' does not exist - I assume the '1' is meant to be an 'l'? Also in line 509.

“619G” is indeed supposed to be 6l9G. We corrected this in the manuscript

Minor points:

L98: The citations given do not specifically refer to encapsulin, although the placement of the citation suggests as much. Please rephrase sentence to ‘isolation of macromolecules from their environment causes loss and change of structural information’ if you would like to retain the citation.

As suggested, we rephrased the sentence to: “As with other proteins, encapsulins function in a complex and crowded cellular environment; isolation of macromolecules from their environment may cause loss and change of structural information [refs].”

L167: ‘The density map we obtained contains protruding densities central in the 5-fold pores.’ This phrasing is somewhat ambiguous as to whether the densities are within the pore or protruding from it (or both).

We rephrased this to: The ‘The density map we obtained contains densities central in the 5-fold pores.’

L201 onwards: You mention in the map that some subunits are ‘partially resolved’. It would be useful to elaborate on what this means on the density level. Does the subunit smear out towards the edge? If so, to which extent, and is it a particular part of the subunit that seems to be poorly resolved? This would be especially helpful since you state that ‘all EncA dimers were intact’ (L198) but also say that an uneven number of monomers were only partially resolved. Perhaps an image of a well-resolved and partially resolved monomer would be helpful for the supplementary material.

To use an objective criterion rather than manual interpretation of the map, we used the cross-correlation scores of the rigid-body refined model to the map of the intermediate encapsulin structures as a metric for whether an Enc monomer in the partial shells is well resolved or not. “Partially resolved” is only defined by a cc-score between 0.5 and 0.75, which does not strictly imply that specific parts of the monomers are less well resolved than others. In practice, this is often the case though, as local resolution generally decreases near the edge of the encapsulin. We added two new panels to Figure 5 to show examples of a monomer with a low cc-score, or “partially resolved” monomer and a better resolved monomer with a higher cross-correlation score, showing both the structure and the density map.

To avoid any confusion with “partially assembled encapsulin” and to not imply that “partially resolved” monomers would have part of the structure poorly resolved, we changed the phrasing to “less well resolved” in the manuscript.

L201: It would also be useful to provide overall RMSDs of the assembly intermediate models compared to the fully assembled shell to quantify the structural similarity between intermediates and the fully assembled shell.

We determined the RMSD values for the intermediate encapsulin structures compared to the structure of the full shell and added these to the results section. Additionally, we added 3 supplementary movies showing the intermediate encapsulin structures with the residues coloured according to the RMSD values with the full shell structure (Supplementary Movie 1-3).

L310: ‘AI protein-folding’ -> ‘AI-based structural prediction’

L310: ‘the AlphaFold Mania [...] has managed to address several of the initial limitations [...]’ -> un-scientific, please rephrase

L314: ‘[...] use smartly chosen subsets [...]’ -> colloquial, please rephrase

Based on the comments from the other reviewers, we removed this section from the manuscript

There is also a number of spelling and grammatical errors and inconsistencies in the manuscript, which distract a bit from the science. I point out the ones that I found below in a separate section but recommend to the authors to please double-check the manuscript.

Spelling errors:

L43: sub-unit -> subunit (spelled correctly in other places)

L124: do not colocalised -> do not colocalise

L143: ‘we isolated E. coli overexpressed encapsulin from M. tuberculosis’: this is grammatically ambiguous, better: ‘we overexpressed and isolated M. tuberculosis encapsulin from E. coli’

L150: ‘We, used 89.072 particles’ -> ‘We used 89,072 particles’. Also, it would be good to keep consistent within the manuscript whether or not you delimit large numbers (i.e., use a comma), which you sometimes do and sometimes do not.

L154: ‘The icosahedral assemble of 60 copies of EncAtb, can be described to consists of [...]’ -> ‘The icosahedral assembly of 60 copies of EncAtb can be described to consist of [...]’

L175: ‘(rmsd)’ -> ‘(RMSD)’

L182: ‘T1 encapsulins’ -> ‘T=1 encapsulins’

L239: ‘as it’s’ -> ‘as it is’

L262+264: ‘M. Hassaicum’ -> ‘M. hassaicum’

L263: M. Tuberculosis’ -> ‘M. tuberculosis’ (correct in other places)

L288: ‘Because all EncA monomers in the intermediate encapsulin structure are present in a

dimer' -> 'as a dimer'

L320: 'Developments for improving SPA processing of structural heterogeneous samples combined with the rapid pace of developments in in situ structural biology, harbour opportunities to understand the structural landscape of a wide range of macromolecules both in context of, and in relation to cellular function.' -> 'Developments for improving SPA processing of structural heterogeneous samples, combined with the rapid pace of developments in in situ structural biology, harbour opportunities to understand the structural landscape of a wide range of macromolecules both in context of, and in relation to, cellular function.'

L442: pdb -> PDB

L444: real space refinement -> real-space refinement

L448: cross correlation -> cross-correlation (correct in other instances)

L456: Clustal -> Clustal Omega

L491: 'We identified a distinct circular cargo in Some' -> 'in some'

L493: 'The shell of most encapsulins appears to be continues' -> 'continuous'

L497: Please have capitalisation after A), B) etc. consistent

L502: Selection of the Atomic model ribbon view -> atomic not capital

L508: Scalebar:25 Å -> space missing

L546: Supplementary figure 2: Please capitalise 'figure' to make consistent with S1

We corrected all of the above and thoroughly checked the manuscript for any other spelling, grammar or consistency errors.

Reviewer #2 (Remarks to the Author):

In this paper, Berger et. al. showed in vitro structure of the fully assembled and three partially assembled encapsulin shells of *Mycobacterium tuberculosis* EncA. The maps obtained by cryoEM single particle analysis reached 2.4 Å resolution for the fully assembled shell, and 4 to 6 Å for the partially assembled shell. Comparing these structures, the authors proposed a model of encapsulin self-assembly from dimers and starting from one side of the shell to the other.

Berger et. al. also presented study of in situ encapsulin structures from cryo electron tomography (CET) on cryo-FIB-lamellae of *Mycobacterium marinum*, showing diverse encapsulins as complete or incomplete icosahedral particles, with various cargo presence and type. Flexible ribbon-like attachments on the exterior of the shell were observed, though their identity and/or function are unknown. Subtomogram-averaging of the encapsulin particles in situ results in a low resolution 3 nm map, possibly because: 1. the magnification used in CET is low with pixel size 4.24 Å; 2. the tilt scheme used is the traditional bi-direction, instead of dose-symmetric; and 3. low number of total particles for the average.

We thank the reviewer for their comments.

Minor points:

1. Line 243 – 248 on page 8: “We observed flexible tail-like extensions on isolated encapsulins, not present on isolates and, to our knowledge, not described in previous studies. This raises the question which protein(s) could form these tail structures, and what could be their function. We consider it unlikely that these extensions are formed from EncA subunits in the process of assembly or disassembly, based on their flexible appearance, their appearance on fully assembled encapsulins, different contrast from the encapsulin shell and that these extensions have not previously been observed in isolated encapsulins.”

a. Were the tail-like extensions observed on isolated encapsulins or not? The description here is inconsistent.

We only observed the tails on intracellular encapsulin, and not on isolates. We corrected the sentence to: “We observed flexible tail-like extensions on intracellular encapsulins.”

b. The reasons why it is not from EncA subunits in the process of assembly or disassembly are unclear or weak:

- i. What does “flexible appearance” mean? In terms of length, shape, or contrast?
- ii. Were they not observed on partially assembled encapsulins? It seems that some are present in Figure 2D, e.g. in the 2nd, 4th, 6th images (please see attached screenshot).
- iii. The “tail” could be a totally different structure made of EncA subunits, compared to the icosahedral shell structure, hence the difference in contrast or image intensity.
- iv. The loss and change of structural information in isolated proteins could explain why the extensions were never observed previously in vitro, even if the extensions were made of the same EncA subunits.

Based on comments from the other reviewers, we removed the discussion point on whether the tails could be formed by Enc, and now only state the scientific need to elucidate their identity.

2. Line 310 – 316 on page 10: “Over the last year, the AlphaFold Mania (Callaway, 2022) has managed to address several of the initial limitations listed in July 2021; the simultaneous design of sequence and structure (Anishchenko et al., 2021; Lovelock et al., 2022; Lutomski et al., 2022), enrichment with ligands and co-factors (Hekkelman et al., 2021), Small (Park et al., 2021) as well as ultra-large super- molecular complexes (Eisenstein, 2021; Callaway, 2022; Mosalaganti et al., 2022) and to use smartly chosen subsets of the PDB to train neural networks to predict distinct conformational states of transporters and receptors (del Alamo et al., 2022).”

This account of recent developments in using machine learning in protein structure prediction seems irrelevant and distracting to the points of the paper.

We removed this section from the manuscript.

Reviewer #3 (Remarks to the Author):

Summary and General Assessment:

In the manuscript “In situ and in vitro cryo-EM reveal structures of mycobacterial encapsulin assembly intermediates”, the authors use cryo-ET and cryo-EM approaches to investigate the assembly dynamics and intermediate states of Mycobacterial encapsulin shells.

This study tackles an interesting and challenging problem regarding the assembly mechanism of encapsulin protein shells which has remained mostly elusive up till now. Some novel and interesting data are presented to support the hypothesis that icosahedral shell formation proceeds via the successive addition of shell protein dimers. However, it appears that the authors are not that familiar with the encapsulin field in general and make some inferences and statements that need to be corrected. Further, a number of inferences and statements are at best speculative and not supported by enough solid data. I am suggesting a number of experiments and changes to the interpretation/text that should be strongly considered by the authors. After all the points listed below have been addressed, I can recommend this study for publication in Communications Biology.

We thank the reviewer for their comments.

- The authors seem to refer to any encapsulin shell protein as EncA. This is incorrect, EncA specifically refer to the encapsulin shell protein encoded in the Myxococcus xanthus FLP encapsulin operon.

We now refer to the encapsulin shell protein in the manuscript using Enc instead of EncA.

- Page 3, line 82: This statement is most likely incorrect. BfrB and FolB were shown to be to some extent encapsulated upon heterologous overexpression with the encapsulin shell protein. However, their purported targeting peptides (TPs) are most likely not true TPs but simply C-terminal extensions very often found in proteins as they deviate substantially from all other known TPs. Further, they are not part of the respective encapsulin operon. In addition, native isolation of the highly homologous Mycobacterium smegmatis DyP

encapsulin definitively showed that only the expected DyP cargo encoded in the encapsulin operon is encapsulated, not BfrB or FolB.

We agree that based on the tomography data, we cannot unambiguously assign the identity of the ring-shaped cargoes and modified the manuscript accordingly. In the results section we now no longer mention FolB as a candidate for its identity, and now only speculate on it in the discussion:

“We found that intracellular encapsulin have cargoes heterogenous in appearance, including distinct ring-shaped cargoes with an outer diameter of 6.9 nm and encapsulins that appear empty. The mycobacterial encapsulin cargoes BfrB, DyP and FolB all have a central ring shape, with outer diameters of 12 nm, 8 nm and 7 nm respectively (Goulding et al., 2005; Gijssbers et al., 2021; Tang et al., 2021). This suggests that DyP and FolB are possible candidates for the cargo’s identity, but based on its size and appearance alone we cannot unambiguously assign its identity.”

Whether BfrB and FolB are encapsulated under any culture conditions at native expression levels, or only when overexpressed (Contreras *et al.* 2014) and whether they should be considered native mycobacterial cargoes is an interesting topic that we agree should be further elucidated and discussed in future studies. We modified the introduction to describe BfrB and FolB as possible secondary cargoes.

- Page 4, line 123: what is the observed average number and range of encapsulins per cell?

We agree that this is an important metric, but there are several difficulties with providing an accurate number from tomography data acquired on FIB/SEM lamellae. The number of encapsulin and bacteria in the tomograms can be easily counted, but this does not allow us to determine the average number of encapsulin per bacteria, because bacteria are never completely in the field of view of a tomogram. Because all tomograms are collected on 100-300 nm thin lamellae, only parts of each bacterium in a tomogram are present in Z, and because of the limited field of view of each tomogram, many bacteria are also only partially present in the tomogram in XY.

As dividing the number of encapsulin we identified by the number of bacteria (partially) present in each tomogram would be much lower than the actual number of encapsulin per bacteria, we prefer not to report this number.

- Page 4, line 133: Again, FolB is not a well-known encapsulin cargo. The observed cargo is very likely the co-encoded DyP found in almost all Mycobacterial encapsulin systems. DyP forms a ring-shaped hexamer of the observed dimensions.

Please see our previous comment on this issue

- Page 4, line 135: what does "many" mean, how many precisely?

We added the number of encapsulin we could identify with one or more tails in the results section, and the caption of Figure 2.

Could this be an artifact?

Based on our experience in interpreting *in situ* cryo tomograms and the frequency and clarity of the tails, we consider it highly likely that these are macromolecular densities, rather than artifacts. The bacteria were vitreous, allowing for cryo-EM imaging in a near-native state, avoiding any fixation artifacts. Although cryo-EM imaging is inherently noisy because of the limited electron dose that can be applied to the sample, and the limited tilt-range of tilt-series acquired with tomography give rise to missing-wedge artifacts, we could consistently observe tail densities in XY slices of different tomograms and encapsulins.

We think this is an important observation to share with the encapsulin field. We agree that additional lines of evidence are needed to unambiguously confirm the existence of tails, as well as to determine its identity and function. We therefore altered all sentences on the existence of tails to state that we observed these densities on the encapsulin surface, rather than stating that encapsulin have tails.

Stating that it protrudes from the center of the pentameric vertex is simply speculation and cannot be stated as such based on the presented data/images.

We agree that more evidence is needed to support this statement and removed it from the manuscript.

Were any of these “tails” visible after subtomogram averaging/sorting?

The tails were not visible in the global encapsulin average, as all the encapsulin surface area's with and without tails are aligned and averaged together.

- Page 5, line 146: It is very difficult to draw any conclusions from an experiment like this carried out under non-physiological conditions (pH 3). It is certainly conceivable, if not likely, that the low pH might have altered the assembly mechanism. This should be taken into account, clearly pointed out, and discussed. This of course means that all data based on this experiment has to be considered in the context of non-physiological conditions. Any statements about assembly mechanisms or intermediates need to be qualified with this in mind.

We now address this point to the discussion on the partial encapsulin structures:

“A potential downside of this approach is the use of low pH to disassemble encapsulin shells, followed by reassembly at neutral pH, for which the assembly and disassembly mechanism could differ compared to intracellular conditions. However, the high resolution of the structures obtained in this study, and the high structural homology to encapsulin structures obtained without pH-dependant dis- and reassembly support the previous finding that this process is reversible (Rahmanpour and Bugg, 2013; Cassidy-Amstutz et al., 2016; Künzle et al., 2018).”

- The authors should carry out SEC and DLS analysis of low pH samples, the proposed mechanism – based on dimer addition to growing shells – implies that encapsulin shell dimers should be present in solution which should be detectable using the methods mentioned above.

In the manuscript we describe a model for encapsulin assembly where either two dimers or two monomers that rapidly dimerise upon binding to the growing encapsulin shell is favoured over inclusion of trimers or pentamers. This proposal is based on:

- 1) The finding that all Enc subunits in the three encapsulin structures of intermediate states are paired via their dimeric interface.
- 2) Multiple trimers and pentamers have one or more Enc subunit missing.
- 3) *In silico* predictions of binding energy for the dimeric, trimeric and pentameric interface in assembled encapsulin shells suggest that the dimeric bond is much stronger compared to the trimeric and pentameric interfaces.
 - a. In the revised version of the manuscript, we supplemented these *in silico* findings with AlphaFold predictions of multimers, which consistently favour the dimeric interface for multimerization over the dimeric and pentameric interfaces.

Our model does not necessitate a (detectable) presence of dimers in solution at low pH as:

- 1) In our experiments, encapsulin shells were disassembled at pH2, but allowed to multimerise at neutral pH, closer to conditions present inside bacteria. The existence of dimers in solution at pH2 is therefore not required for the proposed model.
- 2) The proposed model describes the rapid inclusion of two monomers that rapidly dimerise upon binding to the growing shell as an alternative to the direct inclusion of an already formed dimer.

The presence of Enc dimers in *brevibacterium linens* has also previously been demonstrated using native mass spectrometry (Snijder *et al.* 2016, figure 1c). SEC-DLS on encapsulin at low pH, as already performed by others for different species (Putri *et al.* 2017, Figure 1a; Jones *et al.* 2023), can also likely not distinguish dimers from monomers or other small multimers. We agree that more direct evidence for the inclusions of dimers and/or rapid dimerization upon binding of a monomer to the growing shell would strengthen support for the proposed model, but we don't think SEC and DLS of Enc at low pH would provide this.

- Page 6, line 170: What is shown in panel (I) of Fig. 3 is not 2-fold symmetrical and does not look like the 2-fold pore? Please clarify.

The pores shown in panel I are at the interface between two subunits (red arrow in panel A), which are already described in literature (Sutter *et al.*, 2008; Lončar *et al.*, 2020; Giessen 2024). We previously incorrectly described them in the figure caption as present on the twofold symmetry axis, which we now corrected.

We did not find a clear pore at points of twofold symmetry, which we now also state in the results section. We also changed panel I in Figure 3 to show both the two pores at the interface between two subunits, as well as a point of twofold symmetry.

- Page 6, line 190: More details should be provided in the main text on how this analysis was done, also, supporting 2D and 3D classes should be shown in the context of the overall workflow that resulted in the identification of these heterogeneous shells.

We extended our description in the materials and methods on how this analysis was performed and added a new supplementary figure showing a diagram of the classification steps with images of intermediate classification results (Supplementary Figure 3).

- Page 7, line 220: I don't think this statement is supported by the data. What different cargos?

We rephrased this to state that the cargos are heterogeneous (empty encapsulin shells, some encapsulin with a distinct ring density, non-distinct cargos variable in appearance), rather than implying we can visually identify distinct cargos.

Are the flexible extensions/tails real?

We think that this is the case. Also see our reply to "could this be an artifact".

- Page 8, line 228: Again, the authors say "diverse" cargos, but do not present any evidence for the presence of diverse cargos, also, it is most certainly incorrect to state that you observed FolB, it is most likely DyP, the main cargo of the system, however, you really cannot make any definitive statement here based on the available evidence.

We rephrased this to “cargoes heterogenous in appearance”. Regarding FolB and DyP, see our previous comment on this topic.

- If the authors want to make more definitive statements about cargo identity, they should purify the native encapsulin from *M. marinum*, this would allow the definitive identification of all cargo(s) present under the given growth conditions.

In the revised manuscript we no longer make any statements on the identity of the ring-shaped cargoes in the results section and only discuss candidates in the discussion.

- Page 8, line 238: This seems unlikely considering that all so far characterized encapsulin shells seem to very efficiently assemble even in the absence of cargo.

We rephrased this sentence to repeat that encapsulin shells effectively assemble without cargoes:

“Another mechanism could be that binding of cargo proteins to Enc assists in the initiation of encapsulin shell assembly, although it should be noted that Enc can effectively assemble in the absence of cargoes as well.”

- Page 10, line 298: This whole paragraph seems unnecessary and disconnected from the rest of the manuscript,

We removed the discussion on how structures of heterogenous structural states benefit AI-based structure prediction for these heterogenous states, as training data.

Figure 2. Montage of tomographic slices of intracellular native encapsulin. (A) Most encapsulins were found to